# Low-density lipoprotein receptor–related protein 1 (LRP1) as an auxiliary host factor for RNA viruses

Stephanie Devignot[1], Tim Wai Sha[1], Thomas R Burkard[2], Patrick Schmerer[1], Astrid Hagelkruys[2], Ali Mirazimi[3,4], Ulrich Elling[2], Josef M Penninger[2,5], Friedemann Weber[1,6]

**Viruses with an RNA genome are often the cause of zoonotic infections. In order to identify novel pro-viral host cell factors, we screened a haploid insertion–mutagenized mouse embryonic cell library for clones that are resistant to Rift Valley fever virus (RVFV). This screen returned the low-density lipoprotein receptor–related protein 1 (LRP1) as a top hit, a plasma membrane protein involved in a wide variety of cell activities. Inactivation of *LRP1* in human cells reduced RVFV RNA levels already at the attachment and entry stages of infection. Moreover, the role of LRP1 in promoting RVFV infection was dependent on physiological levels of cholesterol and on endocytosis. In the human cell line HuH-7, LRP1 also promoted early infection stages of sandfly fever Sicilian virus and La Crosse virus, but had a minor effect on late infection by vesicular stomatitis virus, whereas encephalomyocarditis virus was entirely LRP1-independent. Moreover, siRNA experiments in human Calu-3 cells demonstrated that also SARS-CoV-2 infection benefitted from LRP1. Thus, we identified LRP1 as a host factor that supports infection by a spectrum of RNA viruses.**

## Introduction

Pandemics, epidemics, and zoonotic spillover infections are often caused by enveloped RNA viruses (Judson et al, 2018; Zhang et al, 2020a; WHO, 2021; Valero-Rello & Sanjuan, 2022). These pathogens contain an RNA genome of positive-sense or negative-sense polarity that is encapsidated by a viral nucleoprotein and surrounded by a lipid bilayer containing transmembrane glycoproteins. Being intracellular parasites with a comparatively small genome, viruses are exploiting cellular functions for basically every step of their replication cycle. For this, they are subjugating a multitude of host factors by interaction with specific viral proteins, as it is exemplified by the large virus–cell protein interactomes, for example, of SARS–coronavirus 2 (SARS-CoV-2) or influenza virus (Stukalov et al, 2021; Chua et al, 2022). Although RNA viruses are phylogenetically very diverse, it is conceivable that they may use overlapping sets of cellular factors.

Rift Valley fever virus (RVFV; family *Phleboviridae*, order *Bunyavirales*) is an emerging zoonotic negative-strand RNA virus (Wright et al, 2019) that is listed by the WHO among the pathogens posing the greatest public health risk (WHO, 2021). Using RVFV as a model, we aimed to identify host cell factors supporting the viral replication cycle. A mutagenized cell library was iteratively screened for clones that acquired resistance to the highly cyto-pathogenic RVFV as an indicative that the affected host gene is essential for infection. As a top-ranking hit emerged the low-density lipoprotein receptor–related protein 1 (LRP1), a large plasma membrane receptor that can bind and internalize more than 40 different ligands (Franchini & Montagnana, 2011; Gonias & Campana, 2014; Wujak et al, 2016; van de Sluis et al, 2017; Actis Dato & Chiabrando, 2018). In subsequent experiments, we found that LRP1 enhances the ability of RVFV to attach to the cell surface and enter the cytoplasm by endocytosis, and was also an auxiliary host factor for several other RNA viruses including the human pathogenic coronavirus SARS-CoV-2.

## Results

### Forward genetic screen for genes supporting RVFV infection

As RVFV is highly cytolytic, we devised a forward genetic screen that is based on the positive selection of cells deficient in pro-viral genes. A genome-wide library of knockout haploid mouse embryonic stem cells (mESCs), derived from parthenogenetic mouse embryos (Elling et al, 2019), was generated by mutagenizing with a retroviral genetrap (Fig 1A) that disrupts genes in a revertible manner (Elling et al, 2017). Altogether, $5 \times 10^8$ cells were mutagenized by transduction with a genetrap retrovirus at an MOI of 0.02

[1]Institute for Virology, FB10-Veterinary Medicine, Justus-Liebig University, Giessen, Germany  [2]Institute of Molecular Biotechnology of the Austrian Academy of Sciences (IMBA), Vienna, Austria  [3]Public Health Agency of Sweden, Solna, Sweden  [4]Department of Laboratory Medicine, Karolinska Institutet, Solna, Sweden  [5]Department of Medical Genetics, Life Sciences Institute, University of British Columbia, Vancouver, Canada  [6]German Centre for Infection Research (DZIF), Partner Site Giessen, Giessen, Germany

Correspondence: josef.penninger@ubc.ca; friedemann.weber@vetmed.uni-giessen.de

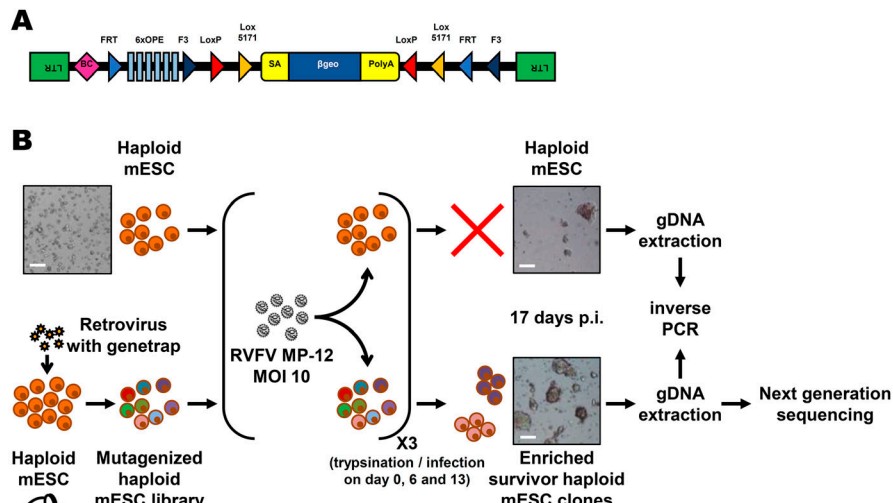

**Figure 1. Forward genetic screen of the haploid mouse embryonic stem cell library for resistance to RVFV MP-12.**
**(A)** Schematic representation of the retroviral revertible genetrap used for mutagenesis. BC, barcode; OPE, Oct4 binding sites; SA, splicing acceptor site.
**(B)** Experimental workflow for the RVFV MP-12 resistance screen using insertional mutagenesis. Bright-field microscopy images of the cells before infection and at the end of the screening process are shown as examples. mESC, mouse embryonic stem cell; p.i, post-infection; RVFV, Rift Valley fever virus. Scale bar: 0.5 mm.

**Table 1. List of clones from the Haplobank collection (haploid mouse embryonic stem cells, AN3-12).**

| Gene | Clone# | Ref. Haplobank | Mutagen | Genetrap orientation | GFP | Cre/mCherry |
|------|--------|----------------|---------|---------------------|-----|-------------|
| LRP1 | 4 | 00800IE06 | retroviral | antisense | wt | KO |
| | 5 | 01128IH08 | transposon | sense | KO | wt |
| | 10 | 00354IA04 | lentiviral | sense | KO | wt |
| | 13 | 00561IB08 | retroviral | sense | KO | wt |
| PTAR | 15 | 01224IG10 | transposon | antisense | wt | KO |
| FBXW11 | 12 | 00371IH05 | lentiviral | sense | KO | wt |
| | 16 | 01258IE05 | transposon | antisense | wt | KO |

resulting in ~$10^7$ independent mutations, selected by neomycin, and expanded. Infection of the parental, non-mutagenized haploid mESCs with the attenuated RVFV strain MP-12 (Ikegami et al, 2015) was productive (Fig S1A) and caused cytopathic effect (CPE) (Fig S1B). Also after infection of the ~75 million genetrap-library cells (7.5 cells/mutation), most of the mutagenized mESCs underwent CPE, but surviving and proliferating cell clones became enriched over 17 d under infection pressure. After repeated cycles of infection and selection (Figs 1B and S2), all the surviving cell clones were collected, and the mutagenized genes were then identified by an inverted PCR/restriction digest assay on genomic cell DNA, followed by next-generation sequencing (Fig S3A and B and Table S1). The top-ranking gene, found to be genetrap-inserted many times independently (Fig S4), was low-density lipoprotein receptor–related protein 1 (*LRP1*). LRP1 is an ~600-kD plasma membrane protein with a 515-kD extracellular part (Franchini & Montagnana, 2011; Gonias & Campana, 2014; Wujak et al, 2016; van de Sluis et al, 2017; Actis Dato & Chiabrando, 2018). For validation, we employed our genome-wide Haplobank library of more than 100,000 haploid mESC clones that contain a revertible genetrap with individual barcodes (Elling et al, 2017). We took a number of independent Haplobank clones that carried a genetrap insertion in an intron of *LRP1* (Table 1). Moreover, Cre/Lox inversion of the genetrap was performed, so each clone

exists in a wt and a knockout version (sister clones; Fig 2A). The mutant or wt mESC clones from the Haplobank and their respective reverted sister clones (each labelled either with GFP or with mCherry, respectively; see Table 1) were then mixed at a 3:7 ratio and subjected to a growth competition assay under RVFV MP-12 infection. Fig 2B shows that in most cases, infected cell clones exhibit a growth advantage when the *LRP1* gene is inactivated. The extent of the growth advantage was different for the different cell clone pairs, but showed a trend towards a certain infection resistance of the *LRP1* knockout cells. For comparison, we employed Haplobank cell clones mutated in previously identified pro-viral host factors of RVFV, namely, prenyltransferase alpha subunit repeat containing 1 (PTAR1; Fig 2C) (Riblett et al, 2016) and the E3 ubiquitin ligase FBXW11 (Fig 2D) (Kainulainen et al, 2016; Mudhasani et al, 2016). Interestingly, in our mESC system the inactivation of these published host factors presented a much weaker survival benefit under selection pressure by RVFV than the deletion of LRP1. Thus, the growth competition experiments were in line with the initial results of the forward genetic screen and indicate that LRP1 may play a role in the life cycle of RVFV. Of note, our screen also returned two other top-scoring genes, *TBX3* and *AIDA*, but subsequent validations showed that unlike *LRP1*, they do not support RVFV replication (Fig S5A and B).

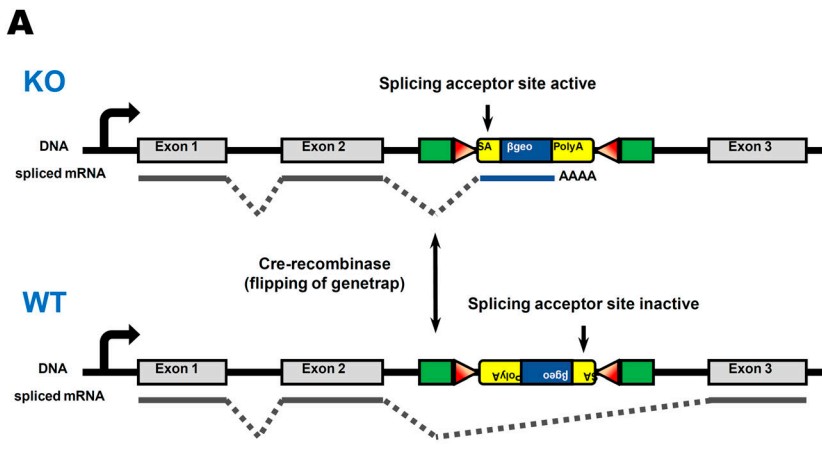

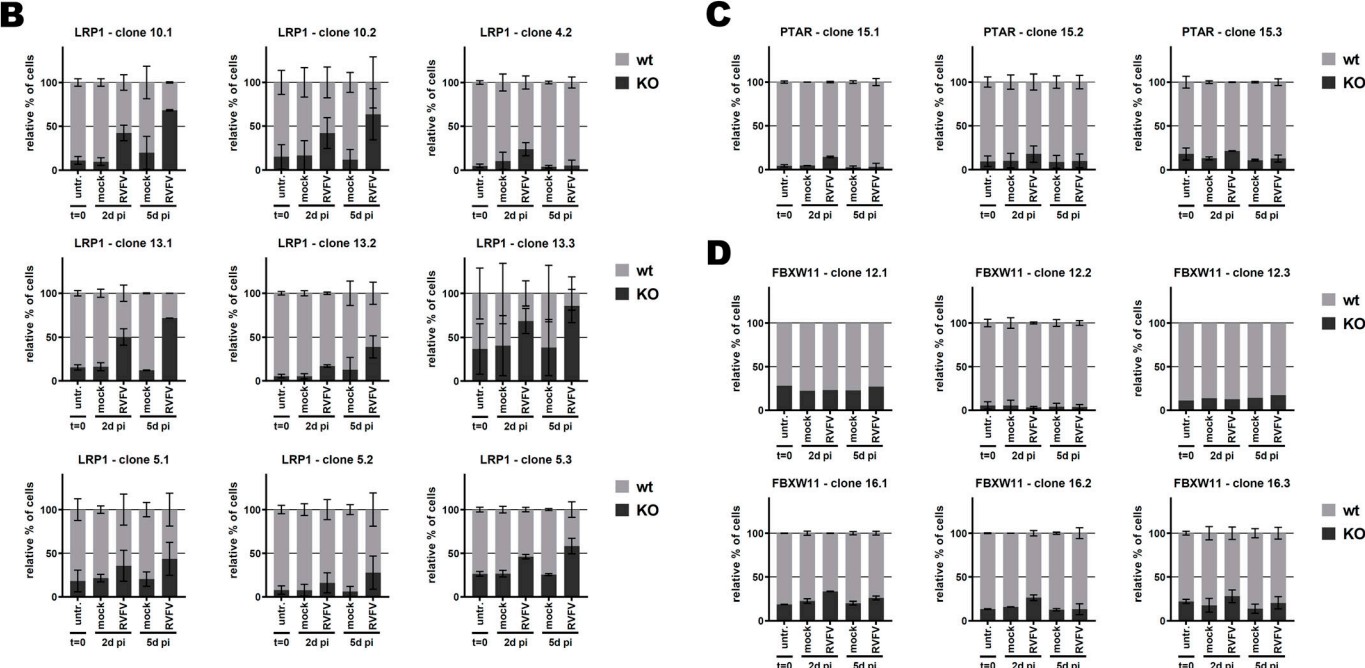

**Figure 2. Growth competition assay in mouse embryonic stem cells.**

**(A)** Schematic representation of the genetrap system when inserted into an intron. When in sense orientation, the genetrap exposes a splicing acceptor site and will be inserted into the mature mRNA, leading to a knockout of the gene of interest. When in antisense orientation, the splicing acceptor site is inactive and the genetrap will be spliced out, leading to a WT expression of the gene of interest. Flipping of the genetrap orientation is possible by expressing the Cre recombinase. **(B, C, D)** Growth competition assay between sister clones bearing a genetrap into the gene of interest ((B): *LRP1*, (C): *PTAR*, and (D): *FBXW11*). Sister clones with WT phenotype are in grey, and sister clones with knockout phenotype are in black (see Table 1). ~30% of knockout cells were mixed with 70% of their WT sister clone, and infected with RVFV MP-12 at an MOI of 5. The ratio between both sister clones was followed by flow cytometry. n = 2, except if the event count was below 1,000, in which case the whole data set was removed (n = 1).

## LRP1 impacts intracellular RNA levels of RVFV, but neither protein synthesis nor particle production

To follow up on our findings from the mouse ESCs, we performed siRNA knockdown of *LRP1* in human A549 cells and tested its effect on RVFV MP-12 replication using RT–qPCR (Bird et al, 2007). siRNA-mediated downmodulation of LRP1 mRNA (Fig S6A) resulted in an up to 50% reduction of RVFV gene expression at 5 and 24 h p.i. (Fig 3A), which is not accompanied by a concomitant increase in cell

survival (Fig S6B). Immunoblot analysis, however, revealed that A549 cells express comparatively little LRP1, whereas in human HuH-7 cells, both the 515-kD alpha chain and the 85-kD beta chain gave strong signals (Fig S6C). Therefore, we robustly down-regulated LRP1 levels in the HuH-7 cells by introducing a CRISPR/Cas9 knockout (Table S2 and Fig S6C), and studied its phenotype with regard to RVFV replication. Also in the HuH-7 *LRP1* knockout cells, RT–qPCR analysis showed suppression of viral gene expression by more than 50% already at 5 p.i. (Fig 3B). However,

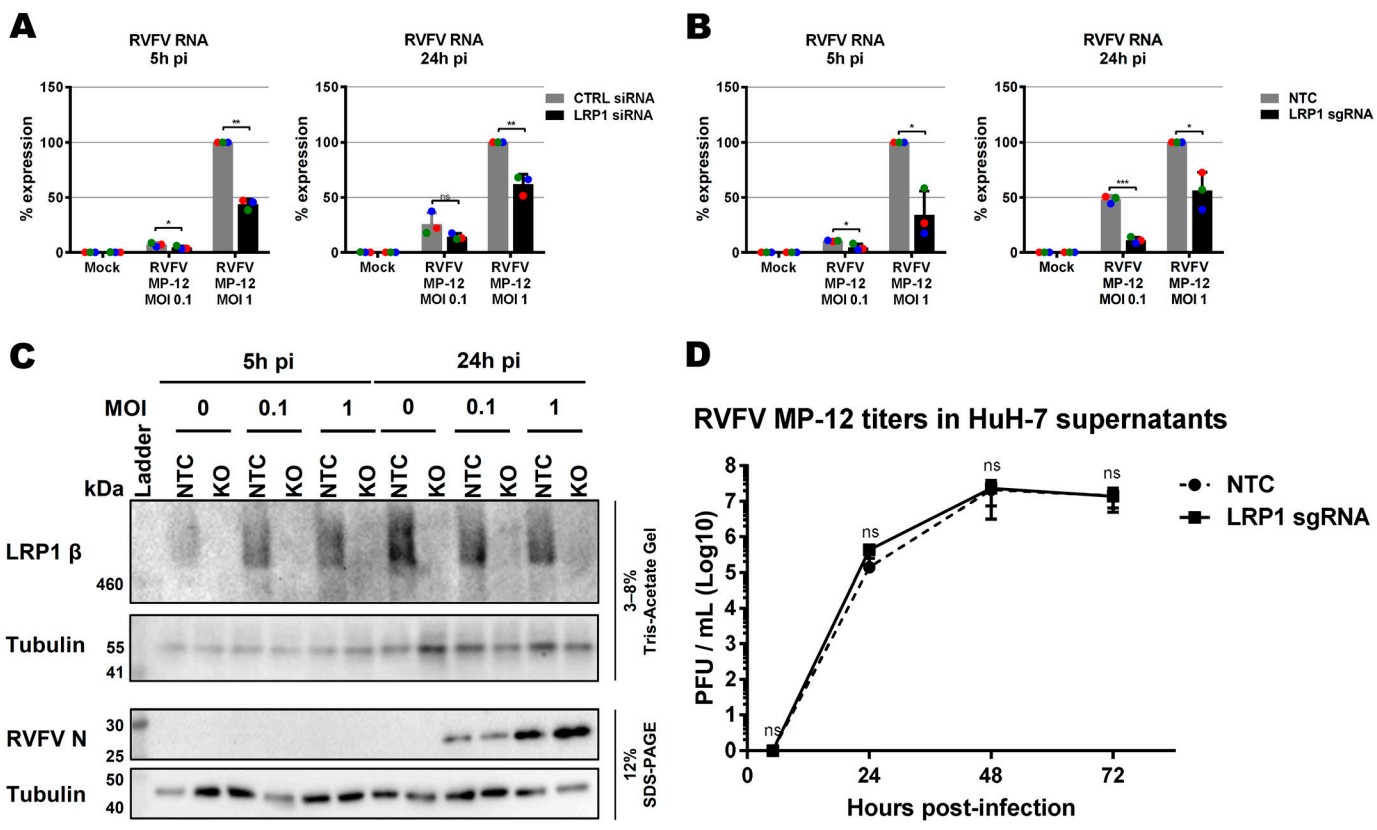

**Figure 3. Influence of LRP1 downmodulation on RVFV MP-12.**
**(A, B)** Virus RNA levels were measured in A549 cell knockdown for *LRP1* (A) and HuH-7 cell knockout for *LRP1* (B). Cells were infected in a synchronized manner with RVFV MP-12 at an MOI of 0.1 or MOI of 1, and RNA was extracted at 5 and 24 h post-infection (p.i.) as indicated. Two-step RT–qPCR was done to detect RVFV MP-12 RNA (L-segment) and the *GAPDH* reference gene. The RNA levels of the RVFV L-segment in the control siRNA or no template control CRISPR/Cas9 cells infected at an MOI of 1 were set to 100%. **(C)** *LRP1* knockout HuH-7 cells were infected with RVFV MP-12 at an MOI of 0.1 or MOI of 1, lysed at 5 and 24 h p.i., and subjected to immunoblotting as indicated. A representative blot is shown. **(D)** HuH-7 CRISPR/Cas9 cells (no template control or *LRP1* knockout) were infected with RVFV MP-12 at an MOI of 0.01, supernatants were harvested at the indicated time points, and infectious virus was measured by the plaque assay. **(D)** Statistics were done on three independent experiments, using a paired one-tailed *t* test ((D): log-transformed data): *, *P* < 0.05; **, *P* < 0.01; ***, *P* < 0.001; and n.s., non-significant.

levels of the viral nucleocapsid (N) protein were unchanged between wt and *LRP1* knockout cells (Figs 3C and S6D). Moreover, when supernatants from cells infected at an MOI of 0.01 were titrated, we did not observe any differences in virus yields (Fig 3D). Thus, the phenotype of LRP1 deficiency under RVFV infection is measurable, but appears to be weaker in human cell lines than in mESCs, possibly because ESCs are in general more prone to virus infection because of their lack of an antiviral interferon system (Guo, 2017). Altogether, we concluded from these data that LRP1 deficiency in human cell lines reduces RNA levels of RVFV, but that its absence seems to have no consequences for the production of the viral N protein or progeny virus particles.

### LRP1 acts early in the RVFV infection cycle

LRP1 is a regulator of cholesterol homeostasis (Lin et al, 2017; Xian et al, 2017), and cholesterol is important for virus infection (Ripa et al, 2021). We therefore investigated whether manipulation of cholesterol levels would influence the phenotype of *LRP1* in RVFV infection. Cholesterol levels were either reduced by using methyl-β-cyclodextrin (MBCD) or increased by enriching the incubation

medium with cholesterol. As shown in Fig 4A, MBCD treatment strongly decreased RVFV infection, as expected, and a difference between LRP1-positive and LRP1-negative cells was not detectable any more. When the cells were given a surplus of cholesterol, infection was slightly reduced in LRP1-positive cells, but not in LRP1-negative cells, and also in this setting, the difference between LRP1-deficient and LRP1-sufficient cells disappeared. Thus, the effect of LRP1 on RVFV infection appears to be dependent on physiological levels of cellular cholesterol.

Like all bunyaviruses, RVFV enters the cells via endocytosis (Albornoz et al, 2016), so we wondered whether the LRP1 effect could be connected to this. Therefore, we either impeded endosomal acidification with bafilomycin A1, or bypassed endocytosis of RVFV particles altogether by acidification of the medium. The inhibition of acidification by bafilomycin A1 strongly impaired infection of the cells, whereas the endocytosis bypass did not substantially affect RVFV RNA levels, but wiped out the difference between the LRP1-positive and LRP1-negative cells (Fig 4B).

Our data suggest that the role of LRP1 in fostering RVFV infection is dependent both on cholesterol and on endocytosis early in infection. To determine the influence of LRP1 on all steps of virus

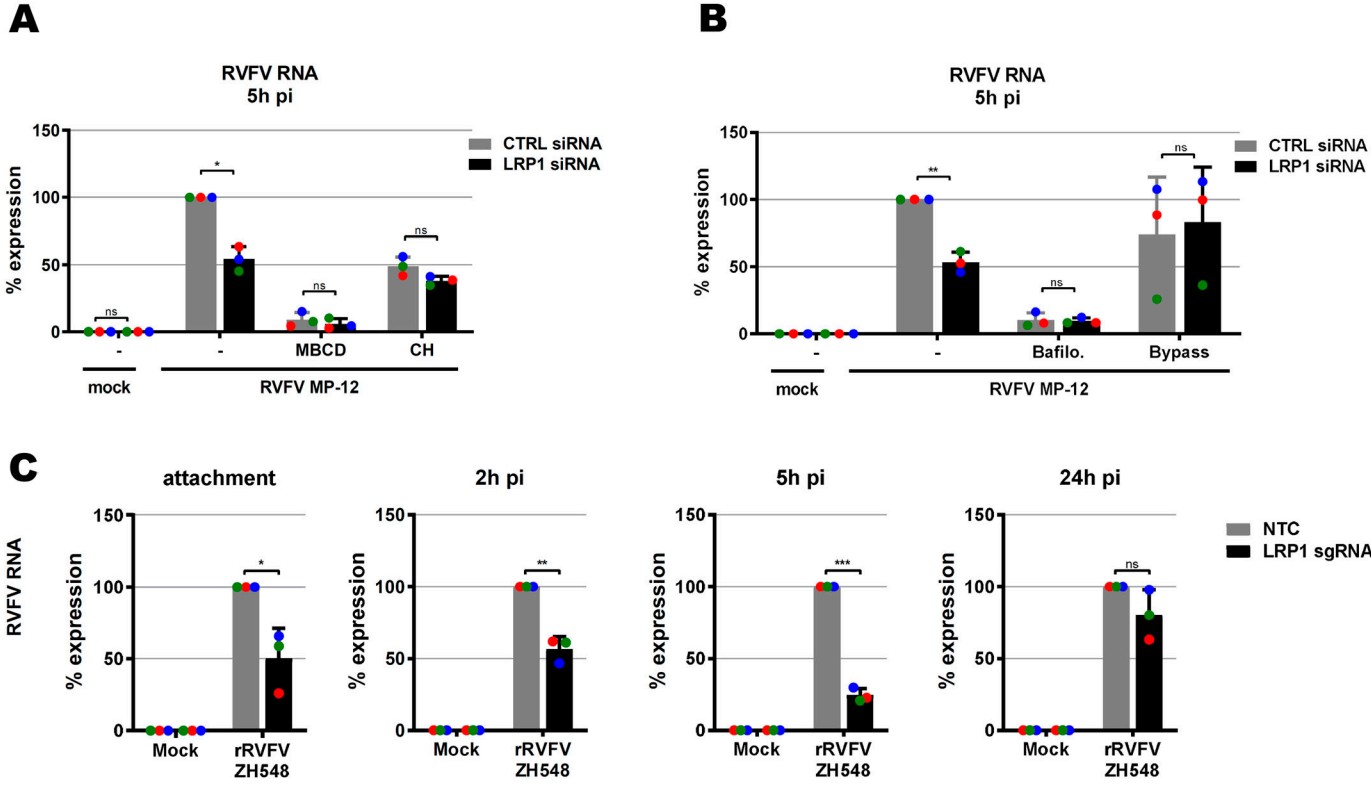

**Figure 4. Mapping the LRP1-promoted infection steps.**
**(A)** Influence of cellular cholesterol levels. *LRP1* siRNA-transfected A549 cells and their controls (see Fig 3A) were pretreated for 1 h with methyl-β-cyclodextrin to deplete cholesterol, or enriched with additional cholesterol (CH), before synchronized infection with RVFV MP-12 at an MOI of 1 for 5 h. **(B)** Role of endocytosis. Cells were pretreated for 1 h with bafilomycin A1 to block endosomal acidification, or incubated for 3 min with an acidic medium (pH 5.0) to force the fusion of viral particles at the cell surface (bypass). **(C)** RVFV ZH548 RNA levels in *LRP1* knockout cells over the course of infection. HuH-7 *LRP1* knockout cells and HuH-7 NTC (no template control) cells were infected in a synchronized manner at an MOI of 1, washed three times, and further incubated in a medium. Samples were collected after the three washes post-infection (attachment step), or at 2, 5, or 24 h post-infection. Two-step RT–qPCR was done to detect viral RNA and the *GAPDH* reference gene. The RNA levels in the infected NTC cells were set to 100%. Statistics were done on three independent experiments, using a paired one-tailed *t* test: *, $P < 0.05$; **, $P < 0.01$; ***, $P < 0.001$; and n.s., non-significant.

replication, we infected HuH-7 wt and *LRP1* knockout cells in a synchronized manner, and took cell-associated RNA samples at 0 h (attachment), 2 h (internalization), 5 h (gene expression after entry into the cytoplasm), and 24 h (late phase of replication). For these analyses, we engaged the wt RVFV strain ZH548, which unlike MP-12 is virulent for humans and animals (Bouloy et al, 2001). As shown in Fig 4C, LRP1 promotes already RVFV particle attachment and internalization, and viral RNA levels in the mutant cells are lagging behind in the subsequent infection stages up to the 5-h p.i. time point. At the 24-h time point, however, ZH548 RNA synthesis has recovered in the *LRP1* knockout cells, different from what was observed with the attenuated mutant strain MP-12 (see Fig 3A and B). Thus, in line with the results from the acidic bypass experiment, this indicates that LRP1 plays a role in the immediate–early steps of RVFV infection, namely, cell attachment and entry. However, despite the differences that LRP1 made regarding viral RNA levels, there was no difference in virus yields (see Fig 3D). It is therefore possible that particle assembly and budding are the rate-limiting steps in RVFV infection, and dominate over the weaker impact that LRP1 has on the immediate–early infection phase.

## LRP1 also supports other RNA viruses

The results from the knockout and time course experiments indicate a role of LRP1 as a cofactor rather than as a main receptor of RVFV infection. We tested its importance for other RNA viruses, namely, the closely related sandfly fever Sicilian virus (SFSV; family *Phleboviridae*, order *Bunyavirales* [Elliott & Brennan, 2014]), the more remotely related La Crosse bunyavirus (LACV; family *Peribunyaviridae*, order *Bunyavirales* [Harding et al, 2019]), and the non-related vesicular stomatitis virus (VSV; family *Rhabdoviridae*, order *Mononegavirales* [Rodriguez et al, 1996]). Moreover, as these all are negative-strand RNA viruses, we also included the non-enveloped, positive-stranded encephalomyocarditis virus (EMCV; family *Picornaviridae* [Carocci & Bakkali-Kassimi, 2012]). As shown in Fig 5A–E, LRP1 was also involved in particle attachment of the bunyaviruses SFSV and LACV. For these viruses, the reduced infection in LRP1-deficient cells was more or less maintained throughout the replication cycle. In contrast, the rhabdovirus VSV and the picornavirus EMCV seem to attach to cells independently of LRP1. For EMCV, none of the replication stages was affected by a lack

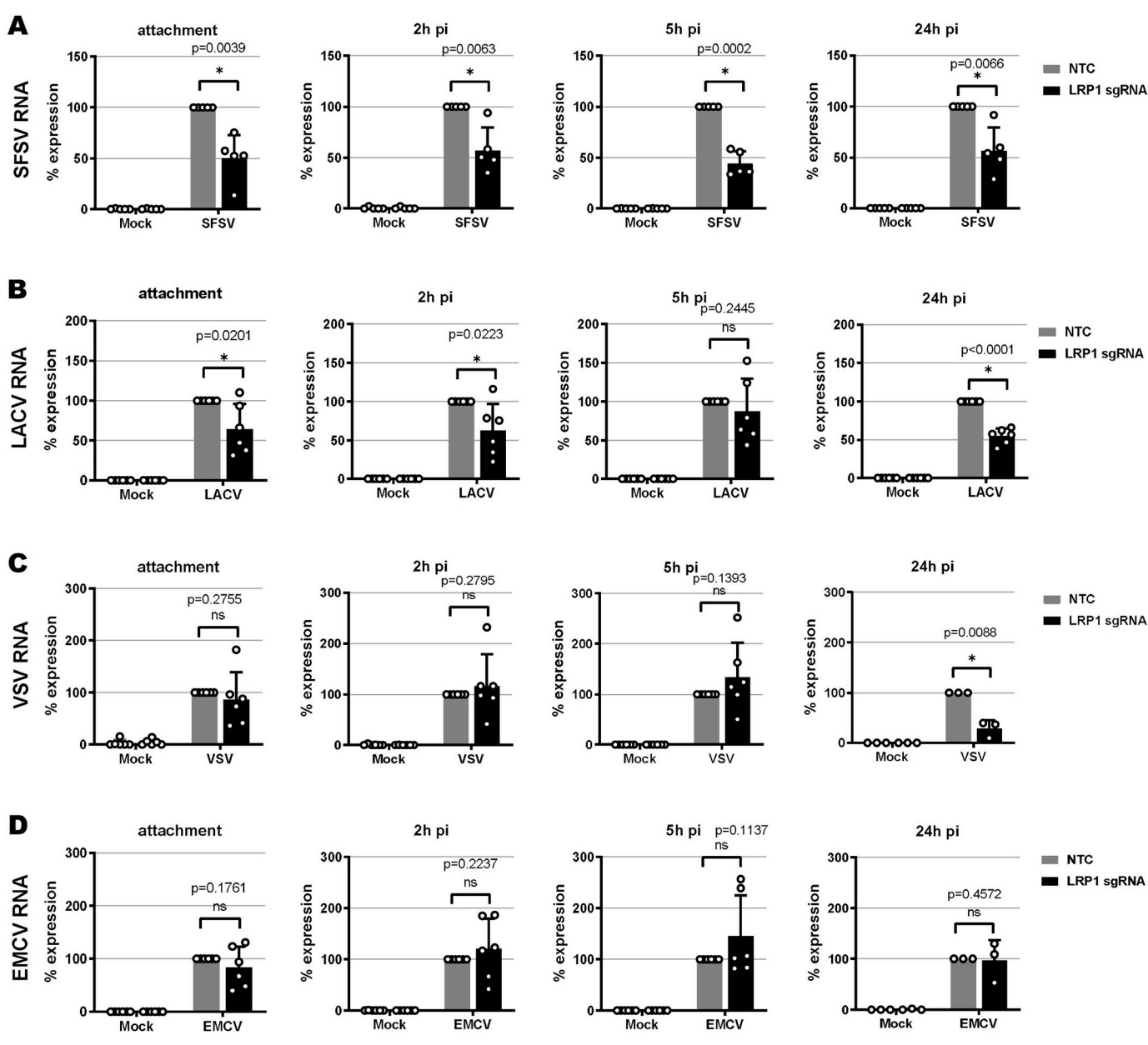

**Figure 5.  Virus RNA levels in *LRP1* knockout cells over the course of infection.**
**(A, B, C, D)** Sandfly fever Sicilian virus, (B) La Crosse virus, (C) vesicular stomatitis virus, and (D) encephalomyocarditis virus. HuH-7 *LRP1* knockout cells and HuH-7 NTC (no template control) cells were infected with the various viruses at an MOI of 1, except for vesicular stomatitis virus that was used at an MOI of 0.1, washed three times, and further incubated in a medium. Samples were collected after the three washes post-infection (attachment step), or at 2, 5, or 24 h post-infection. Two-step RT–qPCR was done to detect viral RNAs, and the *GAPDH* and *18S* rRNA reference genes. The RNA levels in the infected NTC cells were set to 100%. Statistics were done on six independent experiments, using a paired one-tailed *t* test: *, $P < 0.05$; **, $P < 0.01$; ***, $P < 0.001$; and n.s., non-significant.

of LRP1, whereas VSV RNA levels were reduced at the 24-h p.i. time point of infection.

Our comparative time course experiments in HuH-7 cells thus indicate that LRP1 dependency might be a common trait at least for bunyaviruses, but less so for the rhabdovirus VSV and not at all for the picornavirus EMCV. The fact that EMCV infection is unabated at all time points demonstrates that LRP1-deficient cells are in principle still able to support virus infection. For bunyaviruses, LRP1 is facilitating virus attachment, but the overall effect of its absence is comparatively low.

## SARS-CoV-2 infection of cells lacking LRP1

We also investigated the LRP1 dependency of SARS-CoV-2, the causative agent of Coronavirus Disease 2019 (COVID-19) (Coronaviridae Study Group of the International Committee on Taxonomy of Viruses, 2020; Hartenian et al, 2020). For these experiments, we employed the human lung epithelial cell line Calu-3 because our HuH-7 CRISPR/ Cas9 no template control (NTC) and *LRP1* knockout cell clones exhibited differences in levels of the SARS-CoV-2 receptor ACE2

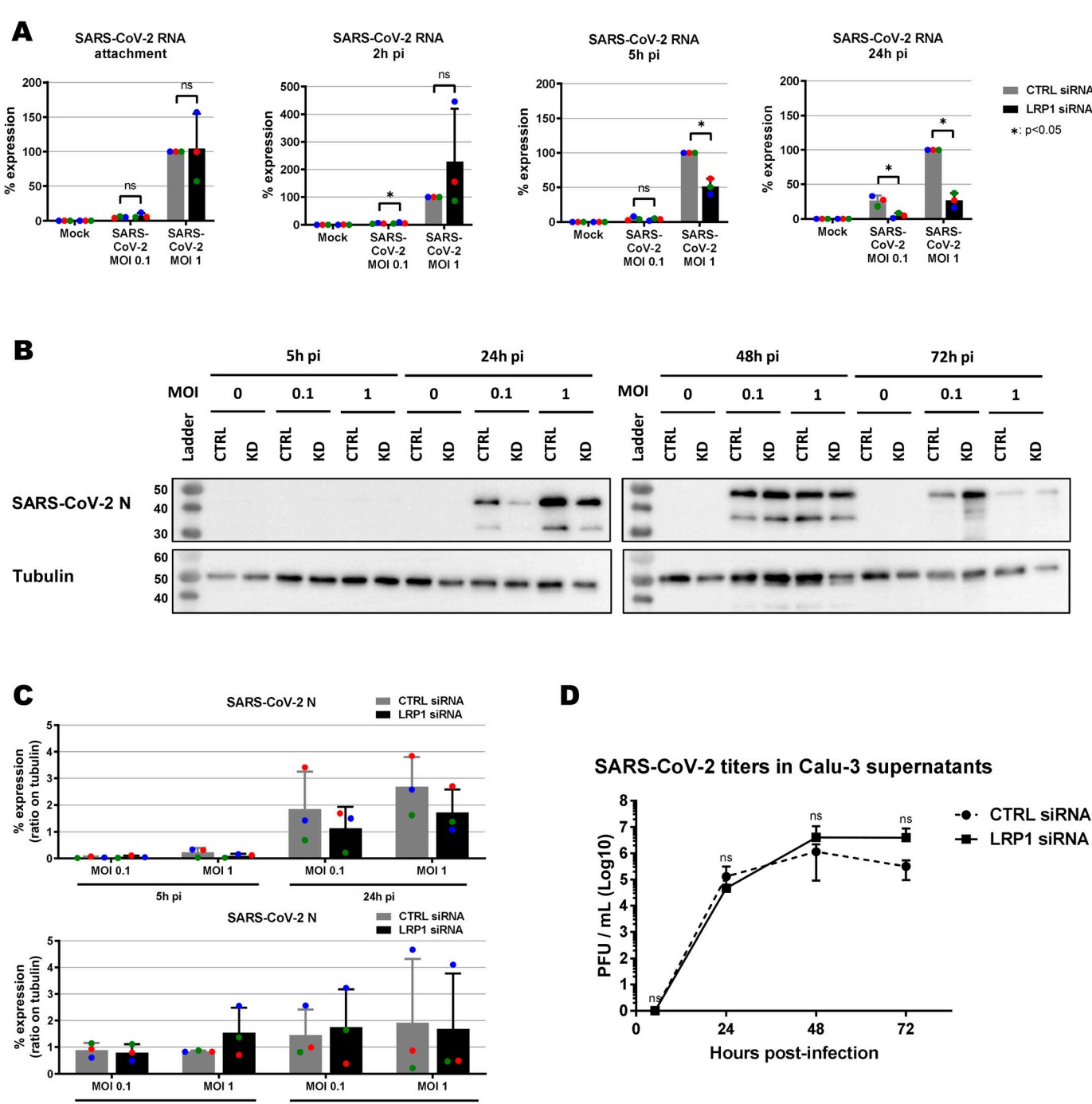

**Figure 6. Effect of *LRP1* knockdown on SARS-CoV-2 multiplication in Calu-3 lung cells.**
**(A, B, C, D)** siRNA-transfected cells were infected at the two indicated MOIs; samples for viral RNA analysis (A), immunoblotting (B, C), and virus yields (MOI 0.01) (D) were taken at the different time points, and analysed as described for Figs 3, 4, and 5. **(B)** Representative blot is shown in (B). **(C)** Quantifications of the N immunoblot signals relative to the tubulin signal are shown in (C). **(D)** Statistics were done on three independent experiments, using a paired one-tailed *t* test ((D): log-transformed data): *, *P* < 0.05; **, *P* < 0.01; ***, *P* < 0.001; and n.s., non-significant.
Source data are available for this figure.

(Fig S6E). When we performed siRNA knockdown in Calu-3 (Fig S7A and B), LRP1 was found to be supporting RNA synthesis at 5 h p.i and 24 h p.i, but not at the earlier stages of the replication cycle (Fig 6A). Moreover, a transient impact on viral N protein synthesis could be

discerned in some of the replicates, but the effect was not statistically significant (Fig 6B and C). In line with this, virus yields were comparable between wt and LRP1-deficient Calu-3 cells at all time points measured (Fig 6D). The positive effect of LRP1 on SARS-CoV-2

infection therefore seems to be transient and limited to the viral RNA levels.

Thus, taken together, our forward genetic screen in haploid mESCs enabled us to identify the cellular protein LRP1 as promoting infection by RVFV and some other RNA viruses including SARS-CoV-2. Although the pro-viral effect of LRP1 was comparatively small, it was measurable, appeared to be independent of the particular cell type, and was mostly applied already in the attachment phase of the replication cycle. Thus, we conclude that LRP1 can act as an auxiliary host factor for enveloped RNA viruses.

# Discussion

LRP1 (or CD91) is a scavenger-type receptor involved in a wide variety of cell activities such as migration, proliferation, differentiation, but also regulation of cholesterol homeostasis, inflammation, or clearance of plasma proteins from the bloodstream (Franchini & Montagnana, 2011; Gonias & Campana, 2014; Wujak et al, 2016; van de Sluis et al, 2017; Actis Dato & Chiabrando, 2018). It is important for the integrity of blood–brain barrier and can bind and internalize more than 40 different ligands (including apoptotic bodies or the Alzheimer's disease–associated tau protein). Moreover, LRP1 modulates signalling pathways, for example, those by JAK/STAT, ERK1/2, or TGF-ß (Wujak et al, 2016; He et al, 2021). Our findings indicate that a series of RNA viruses are aided by LRP1 at their attachment and entry steps, as might be expected from a membrane protein involved in constitutive endocytosis. Moreover, using RVFV as a model, we observe that cholesterol acidification and endosomal acidification are involved in the positive influence of LRP1 on infection. In agreement with these findings, recent studies (published while our article was in preparation) showed that LRP1 acts as a receptor for RVFV and the Oropouche orthobunyavirus by binding to the viral envelope proteins and mediating their entry into the cytoplasm (Ganaie et al, 2021; Schwarz et al, 2022). Strikingly, LRP1 is a transporter that can overcome the blood–brain barrier (Pflanzner et al, 2011). Therefore, it may possibly be involved in the central nervous system tropism that is exhibited by viruses such as RVFV, SFSV, LACV, or Oropouche orthobunyavirus (Alkan et al, 2013; Harding et al, 2019; Chiang et al, 2021; Connors & Hartman, 2022).

For two of our tested viruses—namely VSV and especially SARS-CoV-2—LRP1 appeared to be only relevant for the later stages of infection. However, as the levels of viral RNA are low at the attachment and entry stages of infection, we cannot rule out that for these viruses, the LRP1 effect is too small to be robustly detected at these early stages. Indeed, also Ganaie et al investigated the LRP1 dependency of VSV and observed that the effect of LRP1 on VSV attachment and entry is negligible (Ganaie et al, 2021), but nonetheless a reduction in VSV spread in cell culture was detectable later in the infection cycle (Schwarz et al, 2022). It is therefore conceivable that for some viruses, hard-to-measure minor effects on attachment can amplify in a manner so that they become statistically robust only late in infection. On the contrary, coronaviruses such as SARS-CoV-2 are known to intensively reorganize internal cell membranes in order to generate compartments that can serve as a safe space for transcription and replication of the genome RNA, and to assemble progeny particles (Miller & Krijnse-Locker, 2008; Wolff et al, 2020a; Wolff et al, 2020b). As LRP1 is mostly cycling between the plasma membrane and the endosomes, it may transport factors critical for the RNA replication, for example, lipids that are recruited for the formation of virally induced intracellular membrane compartments. Hence, it appears possible that SARS-CoV-2 indeed engages LRP1 for late-stage replication, whereas for bunyaviruses, it is attachment and entry. Future investigations should clarify whether and how SARS-CoV-2 is engaging LRP1 for its intracellular replication.

The results by us and others (Ganaie et al, 2021; Schwarz et al, 2022) indicate that LRP1 is facilitating attachment and entry for a series of viruses. On the contrary, we found that EMCV is not dependent on LRP1, and Schwarz et al reported the same for the Zika flavivirus (Schwarz et al, 2022). Thus, LRP1 appears to be a broad, but not entirely general, host factor for viruses. As EMCV is non-enveloped but Zika flavivirus is enveloped, the dependency on LRP1 seems not to be due to the presence of a viral lipid envelope.

Our data and those by Ganaie et al (2021) and Schwarz et al (2022) clearly show that LRP1 is the only attachment factor neither for RVFV, nor for any of the other viruses tested. Indeed, heparan sulphates, C-type lectins such as L-SIGN, and the intermediate filament vimentin were identified as auxiliary entry factors for RVFV, the LACV-related Schmallenberg virus, SARS-CoV-2, or many others (de Boer et al, 2012; Gillespie et al, 2016; Leger et al, 2016; Riblett et al, 2016; Cagno et al, 2019; Thamamongood et al, 2020; Zhang et al, 2020; Amraei et al, 2021; Amraei et al, 2022). Interestingly, the low-density lipoprotein receptor, which belongs to the same protein family as LRP1 ("low-density lipoprotein receptor–related protein 1"), is the receptor for VSV (Nikolic et al, 2018) and has been proposed as an entry factor for flaviviruses and hepatitis B virus (Agnello et al, 1999; Krey et al, 2006; Li & Luo, 2021). This may indicate that such endocytosis-active host factors are often exploited by viruses.

# Materials and Methods

### Cells and viruses

A549, BHK, HuH-7, Vero E6, Vero B4, and Calu-3 (kindly provided by Marcel Müller), and *LRP1* knockout HuH-7 cells were grown in DMEM containing 10% FCS, 2 mM glutamine, 100 U/ml penicillin, and 100 µg/ml streptomycin. Medium and supplements were purchased from Thermo Fisher Scientific. The haploid mouse embryonic stem cells AN3-12 were derived and sorted for their haploidy as described (Elling et al, 2011; Elling et al, 2019). The AN3-12 cells, the AN3-12 genetrap-mutated library, and the derived genetrap-mutated specific cell lines from the Haplobank (Elling et al, 2017) were grown in embryonic stem cell medium containing FCS and mouse leukaemia inhibitory factor (Elling et al, 2017). All AN3-12 cells were cultured in 10-cm cell culture dishes and passaged every second day by trypsinizing and reseeding at 1:10, and the medium was changed every other day.

RVFV strain MP-12, LACV, encephalomyocarditis virus strain FA (EMCV), and VSV were propagated in BHK cells. Recombinant RVFV

(strain ZH548) and severe acute respiratory syndrome–coronavirus 2 (SARS-CoV-2, strain München-1.2/2020/984) were propagated in Vero E6 cells, and the sandfly fever Sicilian virus (SFSV, strain Sabin), in Vero B4 cells. All virus stocks were confirmed to be mycoplasma-free. Infection experiments were done under conditions of either biosafety level 2 (BSL-2: EMCV, LACV, RVFV MP-12, SFSV, and VSV) or biosafety level 3 (BSL-3: RVFV ZH548 and SARS-CoV-2).

## Screening of haploid embryonic stem cell library with RVFV MP-12

We used a haploid mouse embryonic stem cell (mESC) barcoded library (complexity: $9.7 \times 10^6$), mutagenized with the genetrap retrovirus JZ-BC frame 0, 1, 2 (Elling et al, 2017). Thawed cells were grown overnight and seeded into seven 15-cm cell culture dishes at a density of $10 \times 10^6$/dish. 4 h later, cells were washed with PBS and infected with RVFV MP-12 at an MOI of 10. As a control, $3.9 \times 10^6$ genetrap-library cells and AN3-12 parental WT cells were seeded into 10-cm dishes, and infected by RVFV MP-12 at an MOI of 10, or incubated with the according mock supernatant. After 1 h at 37°C, infection was stopped by adding medium on top of the inoculum, and cells were further incubated at 37°C. Every 24 h during the whole screening process, the medium was renewed and cell pictures were taken using bright-field microscopy. At day 6 and day 13 post-infection (p.i.), surviving cells were trypsinized, seeded at the same density for WT and genetrap-library cells, and reinfected 4 h later with RVFV MP-12 at an MOI of 10. At day 17 after the first infection, surviving cells were trypsinized and analysed for genetrap vector integration sites.

## Mapping of genomic genetrap vector integration sites

Experimental details of genomic DNA extraction, restriction digest, ring ligation and inverse PCR (iPCR) with primers located in the genetrap (see Fig S3A and B), and next-generation sequencing of integration sites were described previously (Elling et al, 2019). In short, the cell clones that were resistant to infection with RVFV MP-12 were trypsinized, washed in PBS, and incubated overnight in gDNA lysis buffer (GDLB: 10 mM Tris–HCl, pH 8.0, 5 mM EDTA, 100 mM NaCl, 1% SDS, and freshly added 1 mg/ml Proteinase K). After treatment with 10 µl RNase A (QIAGEN) for 1 h at 37°C, genomic DNA was extracted with phenol/chloroform/isoamyl alcohol and precipitated with isopropanol. Pellets were washed in 70% ethanol and dissolved in Tris–EDTA (TE) buffer, and the amount of DNA was measured using QuantiFluor dsDNA Dye (Promega). An aliquot of 10 µg DNA was digested overnight at 37°C using NlaIII and MseI restriction enzymes (NEB), purified using Sera-Mag SpeedBeads (Thermo Fisher Scientific), and resuspended in TE buffer. After DNA ring ligation using T4 ligase (Roche) at 16°C overnight, DNAs were linearized by SbfI (NEB) for 2 h at 37°C and further purified using Sera-Mag SpeedBeads. The integration site was amplified by iPCR, using KlenTaq polymerase (home-made), the primer DS, and one of the index primers (Table S1): 3 min at 95°C; 36 cycles of 13 s at 95°C, 25 s at 61°C, and 1 min 15 s at 72°C; 5 min at 72°C; and at 12°C until completion. The iPCR products were visualized on an agarose gel, and samples from the retro-library were purified with a QIAGEN gel extraction kit.

Purified iPCR products (genetrap library digested by either MseI or NlaIII) were quantified with a NanoDrop and mixed 1:1 to be combined into one next-generation sequencing flow cell. Raw reads were trimmed to 50 nt and processed as in the NCBI Gene Expression Omnibus entry GSM2227065 (Burkard, 2017). In short, reads were aligned to the genome (mm10) with bowtie (v1.2.2) (Langmead et al, 2009). Insertions of disruptive and undisruptive regions of each gene are summed up (see Fig S4). A binomial test of disruptive insertions against undisruptive regions and against disruptive insertions of a retrovirus input (GSM2227065), respectively, was performed for each gene. Genes were ranked by counts of disruptive insertions (DIs) and were selected with a LOFscore ≤ 1e-20 (loss-of-function score). If the LOFscore equals 0, we remove genes with less than 10 DIs and without insertions in the background.

## Growth competition assay

Genetrap-mutagenized clones from the Haplobank collection (Elling et al, 2017) were thawed, grown in 10-cm dishes, and split in six wells of a 24-well plate. Three of the wells were infected with a MLP-puro-GFP retrovirus, and three were infected with a MLP-mCherry-puro-Cre retrovirus (inducing flipping of the genetrap) (Elling et al, 2017). At 24 h p.i., 1 µg/ml puromycin (Invitrogen) was added, and 5 d later, cells were split and aliquots were frozen. For each gene of interest, cells of one GFP-labelled (original clone) and one mCherry-labelled (flipped sister clone) version were mixed at a ratio of ~30% knockout cells to ~70% WT cells, respectively. At 4 h post-seeding, the mixed cells were washed with DMEM and either mock-infected or infected with RVFV MP-12 at an MOI of 5 for 1 h at 37°C. Embryonic stem cell medium was then added on top, and cells were further incubated at 37°C. Cells were trypsinized at 2 and 5 d pi, and either fixed in 4% PFA for flow cytometry analysis or further grown after seeding in a new 24-well plate. The initial ratios between GFP- and mCherry-labelled cells were confirmed, followed over time by flow cytometry (BD FACS LSR Fortessa, with HTS), and analysed with FlowJo software. Only conditions with more than 1,000 acquired events were taken into account for final analysis.

## siRNA knockdown

A549 or Calu-3 cells were seeded into six-well plates and reverse-transfected with LRP1 siRNAs Hs_LRP1_1 (SI00036190), Hs_LRP1_2 (SI00036197), Hs_LRP1_3 (SI00036204), and Hs_LRP1_5 (SI03109400) (FlexiTube GeneSolution, QIAGEN) using Lipofectamine RNAiMAX Reagent (Life Technologies), according to the supplier's protocol. A second reverse transfection was done 2 d later, and the cells were seeded into 12-well plates before infection on the next day.

## Generation of CRISPR/hSpCas9 knockout of HuH-7 cells

HuH-7 cells with a knockout in genes of interest were generated using the CRISPR/hSpCas9 strategy from the Zhang laboratory (Sanjana et al, 2014; Shalem et al, 2014). For the LRP1 gene, sgRNAs (see Table S2) were designed using online tools (https://www.addgene.org/crispr/; http://www.e-crisp.org/E-CRISP/designcrispr.html). After cloning of the

required plasmids, the lentiviruses expressing either the specific CRISPR/hSpCas9/sgRNAs or the NTC CRISPR/hSpCas9 were generated, and then transduced in triplicates into HuH-7 cells. Clonal cell populations were isolated by limiting dilution, following the Addgene protocol (https://www.addgene.org/protocols/limiting-dilution/). Each single colony was further amplified and screened by Western blot. The HuH-7 NTC control cells (clone E5) and the HuH-7 *LRP1* knockout cells (clone C8) were then used in further experiments.

## Reverse transcription and quantitative PCR (RT–qPCR)

Total RNA was extracted from cell lysates using RNeasy (QIAGEN), and an aliquot of 100 ng was reverse-transcribed with PrimeScript RT Reagent Kit with gDNA Eraser (Takara) and the included primer mix. An aliquot of 10 ng cDNA was used as a template for amplifying sequences of human *GAPDH*, and *LRP1* and VSV with corresponding QuantiTect primers (QIAGEN) and specific primers (Table S3), respectively, and the SYBR Premix Ex Taq (Tli RNaseH Plus) kit (Takara). *RRN18S* was amplified in a similar manner, but with 2 ng cDNA as a template. RNA levels of EMCV, LACV, RVFV, SARS-CoV-2, and SFSV were detected using specific primers and TaqMan probes (Table S3) (Bird et al, 2007; Weidmann et al, 2008; Qin et al, 2018; Corman et al, 2020), and the Premix Ex Taq (probe qPCR) kit (Takara). All PCRs were performed in a StepOne Plus instrument (Applied Biosystems). The values obtained for each gene were normalized against GAPDH mRNA levels (or RRN18S mRNA levels in the case of the VSV RNA) using the threshold cycle (ΔΔCT) method (Livak & Schmittgen, 2001).

## Immunoblot analysis

Cells were washed with PBS and lysed in Tissue Protein Extraction Reagent (Thermo Fisher Scientific) containing protease inhibitors (cOmplete Tablets EASYpack; Roche), according to the supplier's protocol. Samples for immunodetection of LRP1 were mixed with 4× sample buffer (143 mM Tris–HCl, pH 6.8 [Acros], 4.7% SDS [Roth], 28.6% glycerol [Roth], and 4.3 mM bromophenol blue [Roth]), whereas samples for detection of RVFV N or SARS-CoV-2 N also contained 20% beta-mercaptoethanol and were incubated for 10 min at 105°C.

For detection of LRP1, samples were run through Criterion XT Tris-acetate precast gels (3–8% gradient) (Bio-Rad) in XT Tricine buffer (Bio-Rad), for 1 h 20 min at 150 V. Proteins were transferred on an EtOH-activated polyvinylidene fluoride membrane (Millipore) by wet blotting overnight at 40 mA and 4°C, using Tris–glycine transfer buffer (5.8 g/l Tris [Acros], 2 g/l glycine [Roth], and 10% absolute EtOH [Roth]). The LRP1 $\alpha$-chain (515 kD) and $\beta$-chain (85 kD) were detected using antibodies 8G1 (2 $\mu$g/ml) (Merck Millipore) and 5A6 (2 $\mu$g/ml) (Merck Millipore), respectively, and a HRP–conjugated goat anti-mouse antibody (1:20,000) (Thermo Fisher Scientific).

For detection of RVFV N, SARS-CoV-2 N, and ACE2 proteins, the samples were run through a home-made 12% SDS–PAGE for 1 h at 200 V. Proteins were transferred on a MeOH-activated polyvinylidene fluoride membrane (Millipore) by semidry blotting for 1 h at 10 V, using semidry blotting buffer (48 mM Tris [Acros], 39 mM glycine [Roth], 1.3 mM SDS [Roth], and 20% MeOH [Roth]). The RVFV N

was detected by the 10A7 antibody at 1.2 ng/$\mu$l (kindly provided by A. Brun, INIA), and a HRP-conjugated goat anti-mouse antibody (1:20,000) (Thermo Fisher Scientific). The SARS-CoV-2 N was detected by the anti-SARS-CoV nucleocapsid antibody (ref 200-401-A50; Biomol) and a HRP-conjugated goat anti-rabbit antibody (1:20,000) (Thermo Fisher Scientific). Human ACE2 was detected by the AF933-SP goat polyclonal antiserum (1:1,000) (R&D Systems) and a HRP-conjugated donkey anti-goat antibody (1:20,000) (Thermo Fisher Scientific). Loading controls were performed by using an anti-tubulin antibody (1:4,000) (ref ab6046; Abcam) and a HRP-conjugated goat anti-rabbit antibody (1:20,000) (Thermo Fisher Scientific). Immunosignals were visualized using the SuperSignal West Femto kit (Pierce) and a ChemiDoc imaging system (Bio-Rad).

## Drug treatment

For assays involving cholesterol depletion or enrichment, cells were washed once with PBS, and pretreated for 1 h at 37°C either with 5 mM MBCD (stock in ddH2O; Sigma-Aldrich) in OptiMEM buffered with 25 mM Hepes (stock in ddH2O; Sigma-Aldrich), or with 100 $\mu$g/ml cholesterol (stock in ddH2O; Sigma-Aldrich) in OptiMEM, respectively. Cell monolayers were then washed three times in PBS and infected as indicated above. After infection, cells were washed three times in PBS and incubated in a medium.

To manipulate the entry of RVFV particles, cells were pretreated for 1 h at 37°C with 20 nM bafilomycin A1 (stock in DMSO; Calbiochem) to block endocytosis. Cell monolayers were then washed three times in PBS and infected as indicated above. After infection, cells were again washed three times in PBS and incubated in a medium supplemented with bafilomycin A1. To bypass the endocytosis step, cells were infected as indicated above, washed three times with PBS, and incubated for 3 min at 37°C with a prewarmed acidic medium, pH 5.0 (adjusted with HCl 1 M, Roth). The cells were then washed once with PBS and further incubated at 37°C with a normal medium. All samples were collected at 5 h p.i for further analysis.

## Infection time course analysis

Subconfluent cell monolayers were washed once with PBS, incubated for 1 h at 4°C with the respective virus or a mock control, and again washed three times with PBS. Samples for analysis of virus attachment were directly collected by incubating in RLT buffer (QIAGEN) for 10 min at room temperature, resuspension, and storage at 4°C until RNA extraction. For analysis of the subsequent replication steps, the medium was added and the cells were further incubated at 37°C. For the entry step, cells were washed once in PBS at 2 h p.i., incubated in trypsin–EDTA for 10 min (A549 and Calu-3) or 3 min (HuH-7) to remove residual attached viral particles, and then washed three times in PBS with centrifugations (5 min at 10,000*g*). Cell pellets were resuspended in RLT buffer and stored at 4°C. At 5 and 24 h p.i., cells were washed once with PBS, incubated with RLT buffer for 10 min at room temperature, resuspended, and stored at 4°C.

## Virus titration

Supernatants of infected cells were collected and cleared by centrifugation at 400$g$ for 5 min. Serial dilutions were made to infect subconfluent Vero E6 monolayers, and infected cells were then incubated in a medium containing 1.5% Avicel for 3 d. Cells were washed twice in PBS and stained for 10 min with a crystal violet solution (0.75% crystal violet, 3.75% formaldehyde, 20% ethanol, and 1% methanol). Cells were then washed, and plaques were counted. Titres were determined as PFU per ml.

## Statistical analysis

Statistical analyses performed are described in the figure legends.

# Supplementary Information

# Acknowledgements

We are indebted to Christian Drosten and Marcel Müller for kindly providing the SARS-CoV-2 virus and Calu-3 cells, respectively, and to Andreas Leibbrandt, Ellen Wetzel, Manuela Kinzer, and Nicole Schuller for helpful discussions and technical support. Work in the authors' laboratories was funded by the Bundesministerium für Bildung und Forschung (Infect-ERA, grant "ESCential") (F Weber, A Mirazimi, and JM Penninger), the Swedish Research Council (VR; no. 2018-05766) (F Weber, A Mirazimi, and JM Penninger), and the Pandemie Netzwerk Land Hessen (F Weber). Moreover, this project has received funding from the Innovative Medicines Initiative 2 Joint Undertaking under grant agreement no. 101005026 (MAD-CoV-2) (F Weber, A Mirazimi, and JM Penninger). This Joint Undertaking receives support from the European Union's Horizon 2020 research and innovation programme and EFPIA.

## Author Contributions

S Devignot: conceptualization, investigation, visualization, methodology, and writing—review and editing.
TW Sha: investigation and visualization.
TR Burkard: investigation and writing—review and editing.
P Schmerer: investigation.
A Hagelkruys: methodology, project administration, and writing—review and editing.
A Mirazimi: conceptualization, funding acquisition, visualization, methodology, project administration, and writing—review and editing.
U Elling: conceptualization, supervision, visualization, methodology, and writing—review and editing.
JM Penninger: conceptualization, supervision, funding acquisition, visualization, methodology, project administration, and writing—review and editing.
F Weber: conceptualization, supervision, funding acquisition, visualization, methodology, project administration, and writing—original draft, review, and editing.

## Conflict of Interest Statement

The authors declare that they have no conflict of interest.

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
