## [Reviewer comments · Life Science Alliance]

Life Science Alliance

Low Density Lipoprotein Receptor-Related Protein 1 (LRP1) as auxiliary host factor for RNA viruses

Stephanie Devignot, Tim Sha, Thomas Burkard, Patrick Schmerer, Astrid Hagelkruys, Ali Mirazimi, Ulrich Elling, Josef Penninger, and Friedemann Weber

DOI: <https://doi.org/10.26508/lsa.202302005>

Corresponding author(s): Friedemann Weber, Justus-Liebig University Gießen

Review Timeline:

Submission Date:	2023-02-22
Editorial Decision:	2023-03-28
Revision Received:	2023-04-04
Accepted:	2023-04-05

Transaction Report:

Please note that the manuscript was reviewed at Review Commons and these reports were taken into account in the decision-making process at Life Science Alliance.

Manuscript number: RC-2022-01305

Corresponding author(s): Friedemann WEBER

1. General Statement

Dear Dr. Monaco, dear Reviewers,

In the initially submitted version of our manuscript, we observed that in our HuH-7 CRISPR/Cas9 knockout cells for LRP1, infection by SARS-CoV-2 was much more severely impacted than for any of the other viruses we have tested (Fig. 7 of the original manuscript). This included even titer reductions. Our conclusion then was that LRP1 may also be supporting viral RNA synthesis directly, not only the attachment and entry steps as for the other viruses. To follow this up for the manuscript revision, we performed a lot of experiments in response to the reviewers' comments, and also some more that were prompted by our internal discussions.

During the course of these experiments, we recently repeated a control experiment we had done in the very beginning of our LRP1-related SARS-CoV-2 work, namely a Western blot analysis for the receptor ACE2.

Our initial ACE2 blots (September 2021) showed no difference between wt, the non-template control (NTC) HuH-7 and the LRP1 KO (A). However, when we did the experiment again in late 2022 to ensure this once more, our HuH-7 NTC control clone now expressed much higher levels than any other of the cells tested, irrespective of whether they were LRP1 negative or positive (B):

Apparently, during the culturing of the NTC clone the cells upregulated ACE2 for unknown reasons.

Therefore, much to our dismay, we had to realize that all previous and new results regarding SARS-CoV-2 infection of the HuH-7 CRIRP/Cas9 cells were artifacts and had to be discarded. The blot from fig B above is now shown as new Fig. S5E.

Needless to say that we lost a tremendous amount of time on futile experiments. Also, due to this, there was no time left for further MERS-CoV experiments, so we had to remove this virus altogether. We are grateful that the comments and requests by the reviewers have led to the discovery of this artefact. Moreover, the reviews have massively helped to improve our manuscript and to establish that LRP1 is an auxiliary factor that promotes the entry of many viruses.

Reviewer #1 (Evidence, reproducibility and clarity (Required)):

Devignot et al. present their results from a haploid insertional mutagenesis screen that identified LRP1 as an important host dependency factor for Rift Valley Fever Virus. They then show that other viruses, including SARS-CoV-2, appear to benefit from the presence of LRP1, at early stages of infection and possibly at later stages of infection. Overall, the genetic approach used to identify LRP1 in an unbiased manner is appreciated. It is just unfortunate that the authors were scooped by another publication. The point that the authors have to make here is that viruses other than RVFV may also use LRP1 for their own life cycles. However, I find that the authors' presentation of the data is confusing and contradictory at times, and they struggle to present a clear message. By suggesting that LRP1 may serve as an attachment/entry factor for some viruses but may serve during the later stages (virus progeny production) for other viruses would require more extensive, deliberate experiments to test the latter. I would suggest the authors recognize that high variance during experimental repetitions has limited the power of what they can conclude, and as a result, they should simplify the message put forth in this manuscript by focusing on the effects of LRP1 as a receptor/attachment factor for RVFV but also some other viruses. This would be appreciated by readers and would push other labs to figure out how LRP1 promotes attachment/entry (simple role as receptor for virus binding or pleiotropic effects based on cholesterol trafficking disruption)?

Reply:

We thank the reviewer for the constructively critical, but positive and encouraging comments. Yes, it is unfortunate that with respect to the identification of LRP1 as an attachment/entry factor for Rift Valley fever virus by a genome-wide screen, we were scooped by the highly respected groups of Amy Hartman and Gaya Amarasinghe. But as the reviewer is pointing out, we confirm and extend those previous publications by expanding the list of viral LRP1 users, including

SARS-CoV-2. Moreover, we addressed the role of cholesterol and of endocytosis in LRP1 action on RVFV. These data are beyond what was previously known.

With respect to the presentation and description of the data, we have done our best to improve the manuscript, and some more clarity is hopefully also added by the new experiments we have performed.

Major points:

1. Once the authors identify LRP1 through a survival-based selection screen, the authors should then validate their results by showing how knockdown of LRP1 (with siRNA) followed by addition of RVFV leads to increased survival of cells (decrease cytopathic effect observed following infection).

Reply:

As proposed by the reviewer, we measured the survival of wt and LRP1 siRNA-treated cells that were infected with RVFV. Unlike the haploid mouse ESCs, infected human A549 cells did not exhibit a survival advantage when depleted of LRP1. This is in line with the lack of differences in virus yields between wt and LRP1 KO cells that we had observed (see Fig. 3D), and is most likely due to the generally lower impact of an LRP1 deficiency on RVFV infection in human cells compared to the haploid mouse ESCs used for the screen. The new data are shown as Suppl. Fig. S6B and mentioned in the text in line 77 to 78.

2. It would be helpful if the authors could identify a virus that is completely unaffected by the lack of LRP1 at all time points considered (this would help establish that its impact on virus infection stems from facilitating virus attachment versus an indirect effect on cell growth/viability).

Reply:

Upon request by reviewer #3 (see below), we have increased in our pan-viral screen (see Fig. 5) the number of biological replicates to six. With these additional data points most of our initial observations held true. For EMCV, however, all stages of the replication cycle were independent of LRP1. This is different from the earlier observation that EMCV particle attachment is lower in LRP1-deficient cells, which was apparently not statistically robust. For all other stages of the EMCV infection the LRP1 independence was also shown in the original version, and confirmed by the n=6 set-up.

Thus, with EMCV we have a virus control that is completely unaffected by the lack of LRP1 at all time points. We thank the reviewer for the suggestion that this demonstrates the absence of an indirect effect on cell growth or viability when LRP1 is lacking. A sentence regarding this was included in the text in line 141 to 142.

3. Effects at the beginning of the life cycle (attachment, endocytosis, fusion) will also impact later stages, so it is most likely that LRP1 affects the early stages directly. The data in Figure 5

do not support that LRP1 is playing different roles in different virus infections (early versus late). The authors simply have a problem with variation for some of their experimental replicates (namely, in the LACV, VSV and ECMV experiments). Also, since these viruses have different life cycle lengths, it is not appropriate to compare them side by side at these fixed time points. My interpretation is that LRP1 knockdown decreases virus particle attachment, for all viruses tested. End of story. The authors cannot state that, because they see viral RNA differences at later (5 hours, 24 hours) time points that LRP1 must be inhibiting late stage steps of the virus life cycle, such as progeny production. Instead, LRP1 may inhibit attachment of certain viruses more than others, and the ones that are the most inhibited will show the greatest suppression of viral RNA at later time points, simply because attachment at the beginning of the life cycle was most affected. For the viruses that show to be affected at 24 hours but not at attachment or 2 hours, that is probably explained by the high variance between experiments (for LACV for example, 3 of 5 experiments showed strong inhibition of attachment). Since the data are normalized, it is likely that lower RNA levels detected at the earliest time points are responsible for the highly fluctuating variance observed.

Reply:

We agree that the most likely explanation for our observations are immediate early effects on attachment and/or entry, and that even the late-only effect by VSV (Fig. 5C) and SARS-CoV-2 (Fig. 6A) might be due to the higher susceptibility of the early infection stages to variations in RNA levels. We therefore now interpret our data more cautiously and discuss them accordingly (lines 182 to 184).

4. Figure 3C: the discrepant results between Figure 3B and 3C probably has to do with the different sensitivities of the two assays. Western blot analysis of RVFV N protein is not going to be as quantitatively informative as the RNA expression detection by RT-PCR. That said, there is actually a slight reduction in RVFV N protein at 24 h pi at MOI of 0.1. In all cases, however, the effect size is small. A reduction of 50% of RNA expression is small, and that's probably why the authors saw no effect on multi-found infection titers in Huh-7 cells (Figure 3D). These data do not afford the authors with a direct measure of whether LRP1 impacts progeny virus production. The authors need to keep their conclusions simple, and in this case, they should simply report that LRP1 reduces RVFV RNA levels 5-24 hours after infection.

Reply:

Also here we agree and now simplify the conclusion from Fig. 3 to “we concluded from these data that LRP1 deficiency in human cell lines reduces RNA levels of RVFV” (line 89). There is indeed a slight reduction of N levels on the Western blot shown in figure 3C. However, upon quantification of the N signal from 3 independent experiments the effect was not statistically significant. We added the N quantification as new figure S6D

5. The data presented in Figure 7 is the most useful and interesting data in the paper. Since the authors see large effects on SARS-CoV-2 titers in Huh-7 sups from cells knocked out for LRP1,

they should check how cell attachment is affected under these conditions (like was done in Figure 6 in Calu-3 cells). Again, I think it is in the authors' interests to focus the role of LRP1 during attachment/entry, and not late stages.

Reply:

As described in the General Statement at the beginning of this rebuttal letter, we had to discard Fig. 7 entirely because the NTC control HuH-7 CRIRP/Cas9 cells acquired increased ACE2 expression over time.

Minor points:

1. 32 references are not enough. Given the breadth of viruses examined here as well as the technical and biological breadth of the subject matter, many more citations should be added.

Reply:

We indeed need to reference individual statements better. Therefore, as requested, we increased the number of citations to 59 in order to accommodate for the virological, technical and biological breadth of the subject matter.

2. There is no mention of other factors that are thought to serve as receptors or attachment factors for RVFV. This should be reviewed in the Intro/Discussion.

Reply:

We now added a discussion on the role of heparan sulfate and other attachment factors (see lines 189 to 204).

3. The effects of LRP1 on cholesterol homeostasis may be partly or wholly responsible for the observed impact of LRP1 knockdown on virus attachment/entry. Can the authors speculate on how LRP1's impact on cholesterol could contribute to their findings as well as the findings from the competing manuscript?

Reply:

Although the reviewer did not request experimental data, we took up this valuable suggestion and tested the effect on cholesterol level manipulations on RVFV infection and the interplay with LRP1. In addition, we tested the effects of inhibition and bypassing of the endosomal entry step. The results from these new experiments, shown in figures 4 A and B and described in the Results part (lines 94 to 109) show that the LRP1 effect on RVFV infection depends on both cholesterol and endosomal entry. We are grateful for this suggestion.

4. It is too general a statement, and therefore misleading, to state that all RNA viruses are dependent on the common host factors of the cell (line 21).

Reply:

We have toned down this statement by turning it into "...it is conceivable that they may use overlapping sets of cellular factors" (line 24 to 25)

5. Figure 4: the authors should see that LRP1 promotes attachment of RVFV particles to cells, but it is not "required" as they say because there is only a 50% loss of attachment observed.

Reply:

Agreed. "Promotes" describes our observations much better. We have eliminated the "required" from the manuscript.

6. The authors do not state in Figure 5 if the receptors for each virus (for example, DPP4 for MERS-CoV) were introduced into the cells.

Reply:

We did not specifically introduce any of the virus receptors into the cells. As those viruses successfully infected the control cells, we assumed their receptors are expressed. The problem with the differential ACE2 expression was identified (see comment above) and the data from the HuH-7 cells removed.

Reviewer #1 (Significance (Required)):

As someone who studies virus-host interactions, I would say that this work is somewhat but not highly significant, and that is partly due to the fact that a competing manuscript has already shown that LRP1 may serve as a receptor for RVFV. However, the potential significance of this submission is highlighted by the possibility that other viruses may also benefit from LRP1. However, in order to distill and take advantage of that novelty, the authors should simply their message and restrict data presentation to those experiments that can be finely controlled. In some cases, the authors should do additional repeat experiments in order to tame their high variance which is confounding their results and frustrating their attempts to form clear conclusions. Readers in the virology and cell biology field would find this interesting once the manuscript is improved.

Reply:

We thank the reviewer for the positive and realistic assessment of our work. It is certainly unfortunate that we had been scooped by the excellent work of Dr. Hartman, but do hope that we could add even more weight to those claims and further extend the knowledge on LRP1-virus interactions.

Reviewer #2 (Evidence, reproducibility and clarity (Required)):

This manuscript identifies LRP1 as a proviral factor for multiple RNA viruses. A strength of the manuscript is the sophisticated embryonic stem cell screening system employed to identify factors that, when knocked out, allow cells to escape from Rift Valley fever virus-induced cell death. This screen identified LRP1 as the top proviral candidate. However, the article is primarily a collection of disparate experiments and several weaknesses are noted:

1. What other candidates were identified in the screen? Were any of these known to be involved in RVFV infections? This should at least be mentioned.

Reply:

We now mention the two other top hits of the screen, TBX3 and AIDA. Unlike LRP1, however, they did not exhibit a promising phenotype in the growth competition assays and the RVFV RT-qPCRs of siRNA-treated cells. The data on these “hits” are mentioned in the text (line 69 to 70) and added as new supplemental figure S5.

2. Figure 2 is confusing and difficult to interpret. It is also unclear why mixtures of cells were studied rather than simply measuring virus infection percentages or replication in pure populations of KO or reverted (WT) cells.

Reply:

Apologies for the confusion. Measuring virus infections in pure populations of KO or reverted (WT) cells would have been much more time-consuming and material-intensive than counting the number of GFP vs mCherry cells per well. The latter procedure has allowed us to rapidly perform a first validation of the screening results. Measurements of virus infection were used further down the pipeline, and for the two other “top hits” TBX3 and AIDA that we had to abandon, virus RNA levels were determined (see Fig. S5 and reply to comment #1 above) We now simplified Figure 2 by adding more colour and explanations to 2A, and by increasing the size of the graph legends in 2B, C, and D. We also removed the GFP / mCherry cartoon (former Fig. 2B), as these technical details are not really necessary for the understanding (but explained in the Materials and Methods) and may make it difficult to follow the story line.

3. The mechanistic studies with regard to the stage of infection at which LRP1 affects RVFV are also confusing. Many of the graphs report small differences in viral RNA levels, often less than 2-fold changes, which is less than a single Ct value change in the qPCR assay. Further, LRP1 does not seem to affect viral protein or infectious virion production. What then is the significance of this work? Further, how does this reconcile with the initial screen in which LRP1 protected cells from virus-induced death?

Reply:

We agree that an impact on viral RNA synthesis by approximately 50% under LRP1 inactivation is small, and we now emphasize this in the text several times (see also our reply to minor comment 5 by reviewer 1). Our values are however not much different from the results by

Ganaie et al. (DOI: 10.1016/j.cell.2021.09.001, cited by us) who observed reduction of viral RNA by 80% in LRP1 KO cells (see their figure 5A).

The discrepancy between the strong phenotype observed in our initial screen in the embryonic stem cells (ESCs) and the subsequent validation in cell lines experiment could possibly be explained by the absence of an antiviral type I system in ESCs, which makes them in general more prone to virus infection. We added this interesting point to our text (lines 86 to 88).

4. Figure 5 measures effects of LRP1 knockout on several other viruses over a timecourse. However, several of the graphs report triplicate measurements that each give a completely different result (major inhibition, no effect, or enhancement of infection). This results in a lack of statistical significance, but whether there is truly no effect is uncertain.

Reply:

In response to this comment, we now increased the number of biological replicates to six for each of the time courses shown in Figure 5. The results now show that for most viruses tested there except EMCV and VSV there is a reduction of infection already at the attachment state in the absence of LRP1. During the entire infection cycle, EMCV infection was never different between control and LRP1 KO cells, and can therefore serve a negative control (see also our reply to major comment 2 by reviewer 1). VSV showed a reduction in LRP1 KO cells at the 24 h stage only. This actually fits with observations by the Hartman group, who had seen very little influence of LRP1 on VSV attachment or entry (Fig. 5 C and D in Ganaie et al.; <https://doi.org/10.1016/j.cell.2021.09.001>), but nonetheless a reduction of VSV spread in cell culture (Fig. 3A in Schwarz et al.; <https://doi.org/10.1073/pnas.2204706119>).

Thus, the reviewer's valuable proposal indeed clarified a few things, and the VSV results are discussed in the context of the SARS-CoV-2 results (line 180 to 184).

5. Effects of LRP1 on SARS-CoV-2 in lung cells were minimal to none, while there does seem to be an effect in Huh7 cells. Overall, at the conclusion of the paper, I am left wondering about how LRP1 mechanistically affects infection of any of the viruses studied, and whether this occurs in cell types relevant to the individual infections.

Reply:

In the Calu-3 cells that we used for SARS-CoV-2 in addition to the HuH-7 cells, we measured an effect of LRP1 on late-stage virus RNA levels, that was similar in size to the one on the other susceptible viruses. For RVFV, we had additionally used human A549 cells, and the results were in our hands comparable to those from the HuH-7 cells (Fig. 3A and B), and also in line with the results by the Hartman group (<https://doi.org/10.1016/j.cell.2021.09.001>), <https://doi.org/10.1073/pnas.2204706119>). So we assume that the results from the HuH-7 knockout cells are representative. The exception was SARS-CoV-2, because of the unfortunate upregulation of ACE2 in the NTC negative control cells, as outlined in our General Statement

above. The human lung epithelial cells Calu-3, which we also used, are an accepted model for SARS-CoV-2 infections (see e.g. <https://pubmed.ncbi.nlm.nih.gov/35344983/>; <https://pubmed.ncbi.nlm.nih.gov/34982558/>; <https://pubmed.ncbi.nlm.nih.gov/33585804/>; <https://pubmed.ncbi.nlm.nih.gov/34106002/>).

Minor:

1. The article needs more precision in its writing. For example, the first sentences of the abstract and introduction imply that RNA viruses are the primary cause of zoonotic disease. This ignores DNA viruses, as well as all other non-virus microbes.

Reply:

We are happy to take this advice to improve the precision of our writing.

Reviewer #2 (Significance (Required)):

LRP1 was previously identified to be an entry factor for Rift Valley Fever Virus. Some data from this manuscript is consistent with this conclusion. Evidence for proviral roles for LRP1 in other RNA virus infections is also presented.

Reply:

We thank the reviewer for this positive and encouraging comment.

Reviewer #3 (Evidence, reproducibility and clarity (Required)):

Summary: In this study, the authors searched for cell components that promote viral infection and propagation by screening a haploid insertion-mutagenized mouse embryonic cell library for clones that rendered them resistant to the zoonotic Rift Valley fever virus (RVFV). This screen identified the Low-Density Lipoprotein Receptor-Related protein 1 (LRP1, or CD91) as a top hit.

Major comments:

The data given as proof of LRP1 role in virus infection is shown in Fig. 2C. The rather small number of replicates (2) precluded a statistical analysis and hence the validity of this main observation is rather weak.

Reply:

Experiments from figure 2C (now 2B) were not meant to be the proof of the role of LRP1 in RVFV infection, but was rather a refinement of the screening. Moreover, the n=2 referred to the number of replicates in each of the 9 panels, and all 9 panels are independent replicates of the same experiment, namely a growth competition between LRP1 KO cells and their sister clones which were reverted to wt again. To avoid the impression that figure 2C (now 2B) would be the

Full Revision

proof of the role of LRP1 in RVFV infection, we changed our wording in the corresponding text (line 67 to 68). The value of the growth competition experiments for a first assessment of the identified host genes is also illustrated by the fact that the two other genes that ranked after LRP1 (AIDA and Tbx3), failed to show a clear effect in both the growth competition and subsequent siRNA knockdown experiments (new figure S5).

Another concern is the finding that there are major differences among the 9 clones shown in Fig. 2C. For instance, almost no viable cells were seen at 5 days post infection of clone 4.2, whereas clone 13.3 cells were barely infected. The authors need to show immunoblots of LRP1 in the various clones and flipped clones. If there is no difference in the levels of LRP1 among these clones then the interpretation of the results must be modified.

Reply:

As the growth competition assays were meant as a first rough validation of the hits that scored in the mESC screen, we did not perform immunoblot analyses. We modified our interpretations to be more cautious with the conclusions (line 67 to 68).

The authors used CRISPR to knockout LRP1 in both A549 and HuH-7 cells and compared virus RNA levels as percent expression, taking expression levels in WT cells as 100%. They did see a significant reduction in viral mRNA levels at 5 h and 24 h (Fig. 3A, B), but also at time=0 (Fig. 4). Therefore, their conclusion that "LRP1 is important for maintaining intracellular RNA levels of RVFV in human cells" is erroneous. The differences are clearly due to the efficiency of virus attachment and no evidence was presented for a role of LRP1 on RNA levels post virus entry. My conclusion is based also on the independent report that LRP1 is an RVFV receptor (reference 15).

Reply:

The reviewer is right, and we corrected our conclusions.

In spite of LRP1 knockout, the cells were infected, suggesting that RVFV may use additional receptors, as is the case with VSV, which uses various LDLR family members, to which LRP1 belongs. Multiple receptor types may also explain how the virus effectively infects A549 cells, which apparently have a very low level of LRP1 (Fig. S5B). The authors should discuss the possible role of additional RVFV receptors, both in the introduction and the discussion sections.

Reply:

As wished, we discussed a series of additional receptors including LDLR (see last chapter of the Discussion).

Despite the differences in viral RNA there was no difference in virus yield in the culture supernatant at 24-72 h (Fig. 3D). The authors should discuss this unexpected discrepancy between cell-associated viral RNA and virus yields. One possible explanation is that the rate-limiting step of virus production is the assembly or budding processes rather than translation.

Reply:

Good point, we added this to the text (line 121 to 124).

In the following paragraph, the authors state that "The results from the knockout and time-course experiments indicate a role of LRP1 as co-factor rather than as the main receptor of RVFV infection". This conclusion is unacceptable based on the results shown for time=0 (Fig. 4). The apparent "drop" in viral RNA at 5 h in the LRP1 knockout cells (Fig. 4) is misleading due to use of % expression rather than showing actual RNA levels. The authors should present the absolute levels of viral RNA as a function of time and not as relative values that just compare WT vs. KO cells. Figs. 5 and 6A should be similarly corrected.

Reply:

In the immediate-early phases of infection, RNA levels of the attaching and entering virus particles are rather low. Using absolute RNA values measured per sample instead of setting the wt and KO conditions in relation to each other would incorporate additional replicate-to-replicate variation and therefore increase the statistical noise. To our knowledge, using % expression is therefore the usual solution to increase discriminatory power in such experiments. We give here a few examples (that include the Cell and PNAS publications on LRP1 by our competitors) where % infection or relative infection were applied:

- Ganaie et al. (<https://doi.org/10.1016/j.cell.2021.09.001>): Figs. 2B, 2E, 2H, 3C, 3F, 4B, 4G, 4H, 4I, 5 A to E, 5 I to K
- Schwarz et al.; <https://doi.org/10.1073/pnas.2204706119>): Figs. 3A, 4A to C, 5A, 5B
- Zhang et al. (<https://doi.org/10.1016/j.jbc.2022.102511>): Figs. 2A to H, 3, 4D to G, 5C, 5E, 5F
- Yi et al. (<https://doi.org/10.1016/j.celrep.2022.110559>): Figs. 2B to F
- Ugalde et al (<https://doi.org/10.15252/embj.2022110727>): Figs. 2B, 2D, 3A, 3B, 3D
- Tse et al. (<https://doi.org/10.1073/pnas.2118126119>): Figs. 5A, 6A, 6B, 7B, 7C
- Guo et al. (<https://doi.org/10.1128/mbio.02566-22>): Figs. 2A to I, 3H, 4A and B
- Riblett et al. (<https://doi.org/10.1128/JVI.02055-15>): Figs. 2B, 3A and B, 4A to C, 5A and B, 6

- Daly et al (<https://www.science.org/doi/10.1126/science.abd3072>): Figs. 1C to F, 2E, 3B, 3F
- Cantuti-Castelvetri et al. (<https://www.science.org/doi/10.1126/science.abd2985>): Figs. 1A to C, 1E, 2D, 2F, 2H
- Sander et al., (<https://doi.org/10.3389/fphys.2022.805565>): Fig. 5D
- Jocher et al., (<https://doi.org/10.15252/embr.202154305>): Figs. 1C to I, 2A to E, 3C and D

The most interesting observation of this study is the impact of LRP1 knockout on replication of SARS-COV1 and 2 (Figs. 5 F, G). The CRISPR knockout technique is not free of off-target events. Therefore, the authors need to do a comparative RNA-seq of the WT and LRP1 knockout HuH-7 cells to ensure that all genes essential for SARS-COV infection or replication remained intact.

Reply:

Our mentioned immunoblots by which we checked once again for expression of ACE2 in the CRISPR/Cas9 cells is along these lines. And the HuH-7 results regarding SARS-CoVs had to be removed anyway due to the ACE2-overexpression in the NTC HuH-7 controls that emerged over time.

As to the study of SARS-COV2 RNA and virus progeny levels, Fig. 6A should be corrected as mentioned above, In Fig. 6B there is no difference between the N protein in WT vs. the KD cells at 1 MOI at all time points if one corrects for the levels of tubulin (Fig. 6B).

Reply:

Regarding the % expression of figure 6A: we had replied to this point above. Regarding the WB signal of 6B, we now quantified the N protein signals from 3 independent repeats, normalized each to the tubulin signal, and observed indeed that the difference was statistically not significant (although there is a trend). The quantifications are shown as new figure 6C and we weakened our conclusions that were deduced from the WBs (line 151 to 153).

Similarly, The authors show that there is no significant difference in virus titer between the WT and KD cells (Fig. 6C). These results weaken considerably the conclusions from Fig. 5F, G. The authors should discuss these differences.

Reply:

We agree and have weakened our conclusions accordingly (e.g. lines 154 to 155).

Full Revision

The differences in SARS COV2 titers seen in the various HuH-7 clones obtained by CRISPR knockdown of LRP1 (Fig. 7C) are puzzling. The authors should perform immunoblotting of LRP1 from these clones and if LRP1 is indeed absent in all of them as should be expected, then the differences in virus titer between WT and the KD clones are not related to LRP1 presence.

Reply:

As we had removed the HuH-7 data on SARS-CoV-2, this point has become obsolete.

Minor comments: Finally, the Introduction section is written as a summary of the results rather than referring to past literature and presenting a hypothesis. This section must be re-written.

Reply:

We did our best to re-write the introduction. Agreed that an Introduction should not be a summary (although this is often seen these days), but a short wrap-up of the results as kind of a teaser is hard to avoid.

Reviewer #3 (Significance (Required)):

The data presented are weak and need to be supported by additional experiments and presented differently.

This reviewer is an expert in viral receptors

March 28, 2023

RE: Life Science Alliance Manuscript #LSA-2023-02005

Friedemann Weber
Justus-Liebig University Gießen
Institute for Virology
Germany

Dear Dr. Weber,

Thank you for submitting your revised manuscript entitled "Low Density Lipoprotein Receptor-Related Protein 1 (LRP1) as an auxiliary host factor for RNA viruses including SARS-CoV-2". We would be happy to publish your paper in Life Science Alliance pending final revisions necessary to meet our formatting guidelines.

- please upload your main and supplementary figures as single files and add a separate figure legend section to your main manuscript text
- please upload your table files as editable doc or excel files
- please add a running title, a summary blurb, and a category for your manuscript to our system
- please add the Twitter handle of your host institute/organization as well as your own or/and one of the authors in our system
- please use the [10 author names, et al.] format in your references (i.e. limit the author names to the first 10)
- please double-check your figure callouts; you have a callout for Figure S5D, but this isn't in the legend or the figure for Fig. 5; you have a callout for Figure 5 E, but this isn't in the legend or the figure
- please add a figure callout for Figure S6D to your main manuscript text
- the references listed in the Supplemental Material should instead be incorporated into the main Reference list

Figure Check:

- please add scale bars to the microscopy images in Figure 1B
- please add sizes next to the blots in Figure 3C and 6B
- please provide the original blots used to make Figure 6B as Source Data

A. FINAL FILES:

B. MANUSCRIPT ORGANIZATION AND FORMATTING:

Sincerely,

Reviewer #1 (Comments to the Authors (Required)):

The authors have extensively revised the manuscript. In some cases, data have been removed because they could not be reproduced, or additional replicates were added and the authors' interpretations were adjusted. Unfortunately, in the end, the authors have lost much of the perceived novelty and significance of their findings. What remains in this revised manuscript, while supporting that LRP1 positively contributes to various virus infections at attachment and post-attachment stages, does not provide convincing evidence that LRP1 is essential to said viruses. In most experiments, loss of LRP1 results in 2-fold or less reduction in virus infection (and when infectious virus titers were measured, often no impact was observed at all). I think this submission should be published in a virology-specific journal because it will require additional work from several virology labs to work out exactly what LRP1 is doing and how important it might be for a given virus.

Reviewer #3 (Comments to the Authors (Required)):

The revised version of this manuscript is now acceptable for publication.

April 5, 2023

RE: Life Science Alliance Manuscript #LSA-2023-02005R

Prof. Friedemann Weber
Justus-Liebig University Gießen
Institute for Virology
Germany

Dear Dr. Weber,

Thank you for submitting your Research Article entitled "Low Density Lipoprotein Receptor-Related Protein 1 (LRP1) as auxiliary host factor for RNA viruses". It is a pleasure to let you know that your manuscript is now accepted for publication in Life Science Alliance. Congratulations on this interesting work.

DISTRIBUTION OF MATERIALS:

Again, congratulations on a very nice paper. I hope you found the review process to be constructive and are pleased with how the manuscript was handled editorially. We look forward to future exciting submissions from your lab.

Sincerely,
